

# Nest site selection and nutritional provision through excreta: a form of parental care in a tropical endogeic earthworm

Angel I. Ortiz-Ceballos[1], Diana Pérez-Staples[1] and Paulino Pérez-Rodríguez[2]

[1] Instituto de Biotecnología y Ecología Aplicada (INBIOTECA), Universidad Veracruzana, Xalapa, Veracruz, México
[2] Programa de Estadística, Colegio de Postgraduados—Campus Montecillo, Texcoco, Estado de México, México

Corresponding author
Angel I. Ortiz-Ceballos,
angortiz@uv.mx

## ABSTRACT

Nest construction is a common form of parental care in soil organisms. However, it is unknown whether the tropical earthworm *Pontoscolex corethrurus* produces nests in soils with low nutritional quality habitats. Here we studied the reproductive behaviour and nest site selection of *P. corethrurus*, and tested the hypothesis whether *P. corethrurus* produces more cocoons in habitats with low nutritional quality. In bidimensional terrariums we evaluated the combined effect of the nutritional quality of habitat: (Poor Quality Habitat = PQH, Medium Quality Habitat = MQH, High Quality Habitat = HQH) and soil depth (Shallow, Intermediate, Deep) in a factorial $3^2$ design. The number and biomass of cocoons, progeny and the production of internal and external excreta were evaluated. The quality habitat and depth of soil and their interaction had a significant effect on nest site construction and the deposition of internal excreta. *Pontoscolex corethrurus* built a higher amount of nests in the PQH-Intermediate and MQH-Intermediate treatments while more internal excreta were found in the HQH-Intermediate treatment. Offspring biomass was positively associated with internal excreta in the PQH (soil only) and MQH (soil + grass) treatments, suggesting that this could be a form of parental care. Since *P. corethrurus* produces more cocoons in low and medium quality habitats, while produces more internal excreta at high quality habitats, there does not seem to be an association between number of offspring and parental care. We suggest *P. corethrurus* could have two reproductive strategies that act as diversified bet-hedging (do not put all cocoons in one basket) behavior in unpredictable environment, and thus build a higher amount of nests in low and medium quality habitats; and another where they produce more internal excreta as a form of parental care in high quality habitats. Parental care in the form of internal excreta may be particularly important in poor and medium quality habitats where offspring biomass increased with internal excreta. Further research is needed on the ecological conditions that favour the evolution of parental care in earthworms according to their ecological category (epigeic, endogeic and anecic).

## INTRODUCTION

Most animals, including the majority of invertebrates, do not provide any form of care for their offspring (*Smiseth, Kölliker & Royle, 2012*). However, some animals make an effort to increase the survival rate of their progeny by protecting them from predators, lack of food, desiccation and other biotic and abiotic threats (*Clutton-Brock, 1991*; *Smiseth, Kölliker & Royle, 2012*; *Furuichi & Kasuya, 2015*). Mammals and birds provide elaborate forms of care by either one or both parents, including: provision of gametes, oviposition-site selection, nest building and burrowing, egg attendance, egg brooding, viviparity, offspring attendance, offspring brooding, food provision and care even after nutritional independence (*Gardner & Smiseth, 2011*; *Trumbo, 2012*; *Smiseth, Kölliker & Royle, 2012*).

Parental care has been widely studied in avian and mammal species but is also prevalent in certain insects, such as water bugs and dung beetles (*Jeanne, 1996*; *Munguía-Steyer & Macías-Ordoñez, 2007*; *Trumbo, 2012*; *Smiseth, Kölliker & Royle, 2012*). Rigorous, dangerous and competitive environments are conducive to the incidence of parental care (*Mori & Chiba, 2009*). Some soil organisms develop parental care in order to increase the survival of their progeny; for example, in at least 11 families of beetles, ants and termites, parental care seems to be a response to severe environments (*Currie, 2001*; *Muller et al., 2005*; *Mori & Chiba, 2009*; *Smiseth, Kölliker & Royle, 2012*). However, despite the fact that earthworms are among the most ecologically important soil organisms (*Lee, 1985*; *Edwards & Bohlen, 1996*), it is unknown whether they exhibit parental care towards their progeny.

Edaphic (physical, chemical and biological), climatic (soil moisture and temperature) and biological (symbiosis, competition, etc.) factors determine the life history of earthworms (*Lee, 1985*; *Edwards & Bohlen, 1996*). Based on their ecological niche, earthworms have been classified into functional groups (epigeic, endogeic and anecic) that develop different reproductive strategies (r and K) in order to more effectively exploit their edaphoclimatic environment (*Lee, 1985*; *Edwards & Bohlen, 1996*). However, very little is known in terms of earthworm behaviour during the reproductive stage. Various earthworms provide cocoons with a small nutritious package that serves as a food source until the offspring are capable of feeding by themselves, thus increasing the survival of their progeny (*Stephenson, 1930*; *Lee, 1985*; *Edwards & Bohlen, 1996*).

*Pontoscolex corethrurus* is a tropical earthworm of extensive distribution in the tropical regions of the world (*Lavelle et al., 1987*; *Hendrix et al., 2008*). Their populations are concentrated in the upper 10 cm of the soil, but may go deeper into the soil during dry periods (*Lavelle et al., 1987*); thus, it is an endogeic species (polyhumic and mesohumic), since the excreta have greater organic matter content than the surrounding soil (*Lavelle et al., 1987*). Within the different tropical soils it inhabits (from 95% sand to 80% clay; *Buch et al., 2011*), its biological activity positively influences soil fertility and plant growth; thus providing environmental services in both agro and natural ecosystems alike (*Scheu, 2003*; *Van Groenigen et al., 2014*), which has led to it being named the "ecosystem engineer" (*Jones, Lawton & Shachak, 1994*; *Hastings et al., 2007*). However, it is often considered an invasive species since it occupies environments disturbed by anthropogenic activities and can have a negative effect through promoting soil compaction (*Chauvel et al., 1999*). It has

been suggested that the wide distribution of *P. corethrurus* is due to its parthenogenetic reproduction (*Hendrix et al., 2008*), but it may also be the result of parental behaviour that increases offspring survival.

The quality of supplied organic residues is important for enhancing earthworm fertility (*García & Fragoso, 2003*; *Ortiz-Ceballos & Fragoso, 2004*; *Marichal et al., 2012*). For example, *Glossodrilus sikuani* produced 9.9 versus 13 cocoons per adult year in a native herbaceous savanna versus a 17-yr old introduced grass-legume pasture (*Brachairia decumbens* and *Pueraria phaseoloides*), respectively (*Jiménez, 1999*; *Jiménez et al., 1999*). In laboratory studies *P. corethrurus* produced 23.9 cocoons· adult· year$^{-1}$ in a soil-sawdust-legume (*Mucuna pruriens* var. *utilis*) environment (*García & Fragoso, 2003*), while in a natural savannah their fertility varied from 1–15 cocoons· adult· year$^{-1}$ (*Lavelle et al., 1987*).

Construction of nests and providing high quality food (for example, nitrogen in the excreta) are a form of parental care that is common among both vertebrates and invertebrates (*Clutton-Brock, 1991*; *Mori & Chiba, 2009*; *Gardner & Smiseth, 2011*). Previous studies have documented that *P. corethrurus* constructs incubation nests that contain one cocoon per nest and around these they build ''feeding chambers'' where excreta are deposited (*Ortiz-Ceballos & Fragoso, 2006*; *Ortiz-Ceballos, Hernández-García & Galindo-González, 2009*; *Buch et al., 2011*), whereas the anecic earthworm *Lumbricus terrestris* covers its cocoons with its own excreta (*Ramisch & Graff, 1985*; *Grigoropoulou, Butt & Lowe, 2008*). However, despite its acknowledge importance as ecosystem engineer or invasive species, there is presently very scarce information regarding its basic reproductive biology. In particular, it is unclear whether earthworms choose to construct nests and place their cocoons according to habitat quality and how they place their excreta and cocoons when food is either ephemeral and/or distributed irregularly (*Mori & Chiba, 2009*). Here we determined whether *P. corethrurus* produces cocoons based on the nutritional quality of the habitat. We predicted that locations with low quality would demand more parental care, and thus would not be favoured as construction sites. Habitat quality was manipulated by combining different soil depths and nutritional quality.

## MATERIALS AND METHODS

### Terrariums

Fifteen bidimensional (45 × 35 cm) terrariums were utilized in the study. These were constructed of two panes of glass 5.3 mm thick, separated by thin balsa wood strips leaving an internal space of 0.5 cm (*Evans, 1947*; *Capowiez, 2000*; *Ortiz-Ceballos, Hernández-García & Galindo-González, 2009*). The glass was glued to the balsa wood on the two sides and bottom of the terrarium, leaving the top open. The sides of the terrariums were sealed with transparent adhesive tape. Four holes (2 mm wide) were made on the bottom in order to allow water to enter by capillary action.

### Soil

Ten kg of soil were collected from a plot of maize under rotation with the tropical legume velvet bean [*Mucuna pruriens* var. *utilis* (Wall. ex Wight) Back. ex Burck] in the locality of

Tamulté de las Sabanas (18°08′N, 92°47′W), 30 km east of Villahermosa, Tabasco, Mexico. The silty clay loam soil (31.6% silt, 26.8% clay and 41.5% sand) was air-dried in the shade at room temperature, and sieved through a 5 mm mesh. The main chemical characteristics of this soil were: 2.7% organic matter, 0.14% total N, 11.5 C:N; with a pH ($H_2O$) of 6.3. *Pontoscolex corethrurus* was scarce in these plots despite their high abundance a cross this region of Tabasco, Mexico (*Ortiz-Ceballos & Fragoso, 2004*). Previous studies showed that *P. corethrurus* grew and reproduced on soil collected in Tamulte (*Ortiz-Ceballos et al., 2005*; *Ortiz-Ceballos, Hernández-García & Galindo-González, 2009*).

## Earthworms

Thirty subadult *P. corethrurus* earthworms were collected from a pasture of *Brachiaria humidicola* (Rendle) Schweick located at Huimanguillo (17°48′N, 93°28′W), 79 km southwest Villahermosa, Tabasco. The earthworms were reared until reaching sexual maturity in boxes (12 × 12 × 8 cm) with 300 g of the soil mixed with 3% legume (*Mucuna pruriens* var. *utilis*) foliage. Prior to initiation of the experiment, the first 15 earthworms to produce a cocoon were selected.

## Habitat quality

The influence of habitat quality was evaluated using foliage from a legume (*M. pruriens* var. *utilis*) or grass (*B. humidicola*) with 14.3 and 6.1% crude protein, respectively. These were collected from the same sites as the earthworms. 5 kg of legume and grass foliage were collected and dried at 65 °C for 48 h. The dried materials were then sieved to 2 mm and 3.3 kg of the soil was homogeneously mixed with 0.01 kg (3%) of leguminous foliage, while another 3.3 kg of soil was homogeneously mixed with grass (3%).

## Experimental set-up

To test preferences for nest location, an experiment was established utilizing a $3^2$ factorial design, i.e., two factors (food quality and soil depth) with three levels. Nutritional quality of habitat consisted of either: soil only = Poor Quality Habitat (PQH), soil + grass = Medium Quality Habitat (MQH), or soil + legume = High Quality Habitat (HQH). The different soil depths tested were: 0–9 cm (Shallow), 10–18 cm (Intermediate), or 19–27 cm (Deep). Each treatment had five replicates, utilizing a total of 15 terrariums. The terrariums were separated into three depths (layers), each containing 220 g of substrate with the following treatments (Fig. 1): PQH-Shallow, HQH-Intermediate, MQH-Deep, HQH-Shallow, MQH-Intermediate, PQH- Deep, MQH-Shallow, PQH-Intermediate and HQH-Deep. The soil was then moistened with distilled water through capillary action to field capacity with 217 ml per terrarium. As *P. corethrurus* is parthenogenetic, one adult (with clitellum) of similar biomass (455 ± 25 mg) was introduced in each terrarium. The density was equivalent to the abundance and biomass recorded in the field (438 earthworms m$^{-2}$ and 27 gm$^{-2}$, respectively). The terrariums were placed in an incubator at a temperature of 26 ± 1 °C, which falls within the optimal growth and reproduction temperature range (20–30 °C and 23–27 °C, respectively) for this species (*Lavelle et al., 1987*). Water was added through capillary action every six days in order to maintain soil moisture content (measured by the gravimetric method) at water-holding capacity (42%,

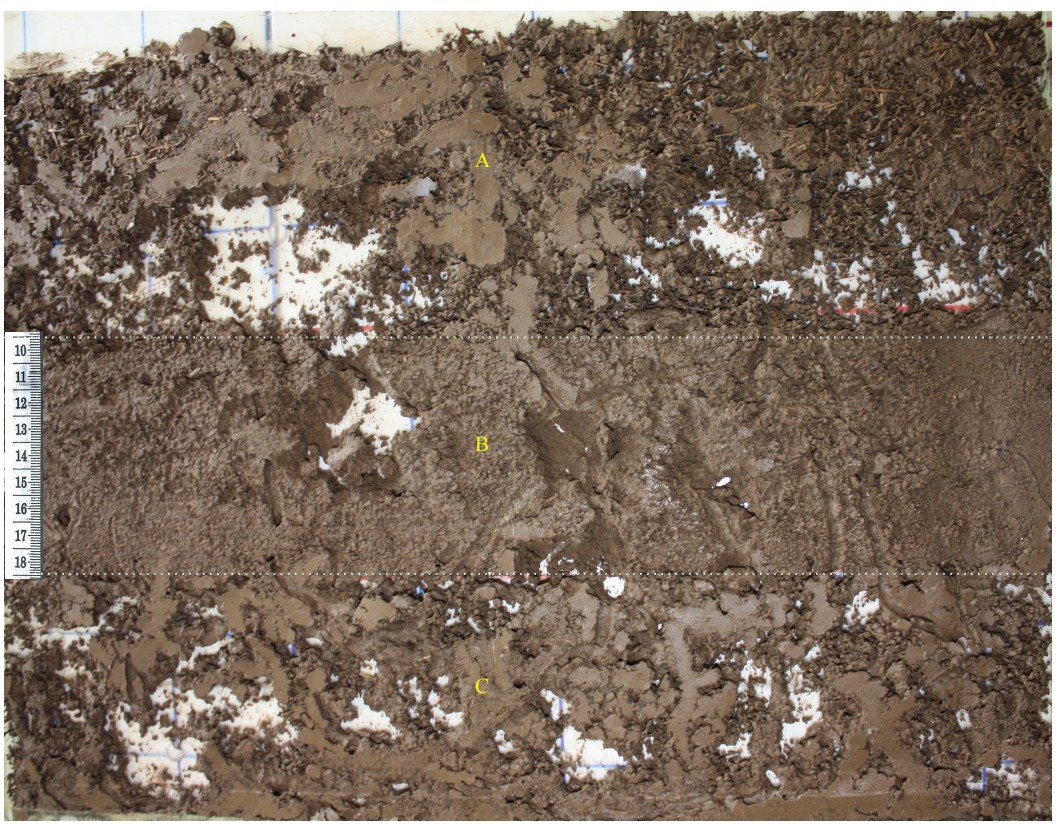

**Figure 1** **Example of the nutritional quality of habitat terrarium with a soil profile.** (A) 0–9, (B) 10–18, and (C) 19–27 cm of the surface (MQH = soil + grass), intermediate (PQH = soil only) and deep (HQH = soil + legume) layer, respectively.

Ortiz-Ceballos, Fragoso & Brown, 2007). Experiments were carried out at INBIOTECA, Universidad Veracruzana, Xalapa, Veracruz, Mexico.

Every third day, cocoon production and emergence of juveniles were recorded. The position of cocoons was marked. After 100 days all terrariums were sampled according to Ortiz-Ceballos, Hernández-García & Galindo-González (2009). The number and biomass of cocoons, juveniles and adults were recorded. One pane of glass from each terrarium was separated to collect earthworms, cocoons and excreta. Juveniles and cocoons were then manually removed from each layer, counted and weighed. In addition, the external (deposited on the soil surface) and internal (within each soil layer) excreta were easy to observe and collected with tweezers, oven-dried (at 65 °C for 72 h) and weighed.

## Data analysis

Biomass parental earthworm (initial and final), total number cocoons, and biomass and number of juveniles by terrarium were compared between treatments by one-way ANOVA for each variable. The analysis was performed using Statistica software, ver 7 (StatSoft, Tulsa, OK, USA).

To analyze the number of cocoons per treatment ($y$) we used a negative Binomial distribution as described in brief below:

$$P\left(Y=y\right) = \frac{\Gamma\left(k^{-1}+y\right)}{\Gamma\left(k^{-1}\right)y!}\left(\frac{k\mu}{1+k\mu}\right)^{y}\left(\frac{k\mu}{1+k\mu}\right)^{1/k}, y = 0,1,2\ldots$$

where: $k$ is a parameter to be estimated, $\log(\mu) = \mathbf{x}'\boldsymbol{\beta}$, which considers the effect of the factors considered in the experiment (soil depth, food quality, etc.). This model was fitted using the GENMOD routine included in the program SAS/STAT for Windows (SAS, Cary, NC, USA).

To analyze the weight of the excreta and its placement (internal or external), we used a linear model, with food type, soil depth and their interaction as independent factors. The model was fitted using the ANOVA routine of the software SAS 9.4 for windows. Means of the treatments were compared using a Tukey's test.

To determine if the biomass of offspring was correlated to the amount of internal excreta we analysed the excreta depostited in the S, G and L treatments using a Pearson's correlation test. Once a relationship was established we fitted a lineal model to find which type of excreta better predicted offspring biomass by comparing the estimated regression coefficients. Analysis were carried out using R software (*R Core Team, 2015*).

## RESULTS

### Nest construction and total number of cocoons laid

*Pontoscolex corethrurus* constructed one nest per cocoon ($0.035 \pm 0.006$ g). There was no significant difference in the weight of the cocoon per treatment ($F_{2,\,72} = 1.26$, $P = 0.29$). Over the experimental period of 100 days, *P. corethrurus* produced an average of $14.6 \pm 3.1$ cocoons per terrarium and $0.505 \pm 0.148$ g per cocoon. There was no significant difference in the number of cocoons produced per terrarium ($F_{2,\,12} = 0.368$, $P = 0.699$). At the end of the experiment, parental earthworms were found to share a similar biomass (average $\pm$ SD) $0.767 \pm 0.13$ g between terrariums. During their reproductive activity the earthworms invested 4.56% of their weight in the formation of a cocoon, which corresponds to 65.84% of the total energy.

### Site selection for nest placement and deposition of cocoons

Habitat quality, soil depth and their interaction had a significant effect on the construction of nests ($F_{2,\,36} = 7.29$, $P = 0.026$ habitat quality, $F_{2,\,36} = 51.42$; $P = 0.0001$ soil depth, $F_{4,\,36} = 14.00$; $P = 0.007$, habitat quality $\times$ soil depth). More nests were found in the intermediate (10–18 cm) soil depth layer and treatments PQH (soil only) and MQH (soil + grass) (Figs. 2 and 3), while treatments HQH (soil + legume) and MQH and the shallow layers (0–9 cm) presented a lower number of nests.

### External and internal deposition of excreta

Earthworms deposited an average of $10.8 \pm 3.9$ g of dry excreta, however, there was no significant difference in the production of superficial (external) excreta between treatments ($F_{2,\,12} = 0.186$, $P = 0.833$). In contrast, the production of internal excreta varied significantly with habitat quality, soil depth and their interaction ($F_{2,\,36} = 21.96$,

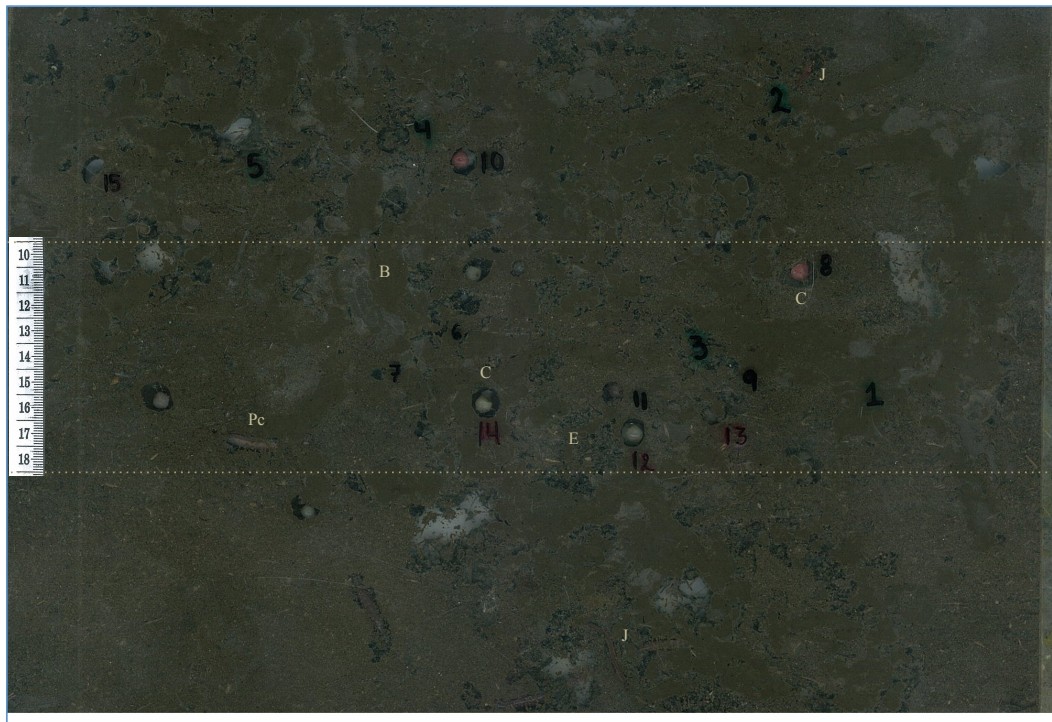

**Figure 2** **Terrarium showing the tropical endogeic earthworm *Pontoscolex corethrurus* (Pc) at the intermediate soil layer where it built a higher number of nests.** Also shown is the burrow system (B), the color (white to pink) of the cocoons (C), indicating the degree of embryo development, offspring (J), and excreta used as food (E).

$P = 0.0001$ habitat quality type; $F_{2, 36} = 4.94, P = 0.0127$ soil depth; $F_{4, 36} = 3.81; P = 0.011$, habitat quality type × soil depth). HQH (soil + legume) and the intermediate soil depth layer had the highest quantity of internal dry excreta (40.80 and 34.85 g), while treatments PQH and the shallow and deep layers (0–9 and 19–27 cm, respectively) had lower quantities of internal dry excreta (16.09, 25.39 and 24.11 g, respectively) (Fig. 4).

### Offspring number and weight

Juveniles weighed on average 9.3 ± 3.1 g per terrarium and 4.85 ± 1.42 g per offspring. There was no significant difference in the number and biomass of juveniles per terrarium between treatments ($F_{2, 12} = 0.75, P = 0.49$ and $F_{2, 12} = 2.85, P = 0.09$, respectively). The Correlation analysis found that the internal excreta deposited in treatments PQH and MQH where positively associated with the biomass of juveniles ($r_{15} = 0.68, P < 0.005$ y $r_{15} = 0.53, P < 0.043$, respectively), but was not associated to the excreta in treatment HQH ($r_{15} = 0.240, P < 0.389$). The internal excreta placed in treatment PQH had a strong association with the biomass of juveniles (Fig. 5) with an estimated regression coefficient of $\hat{\beta} = 0.012 ± 0.003$ ($F_{1, 13} = 11.08, P = 0.005, R^2 = 0.678 ± 0.203$). There was a weaker association between juvenile biomass and the excreta deposited in treatment MQH ($\hat{\beta} = 0.006 ± 0.0027, F_{1, 13} = 5.05, P = 0.043, R^2 = 0.528 ± 0.23$).

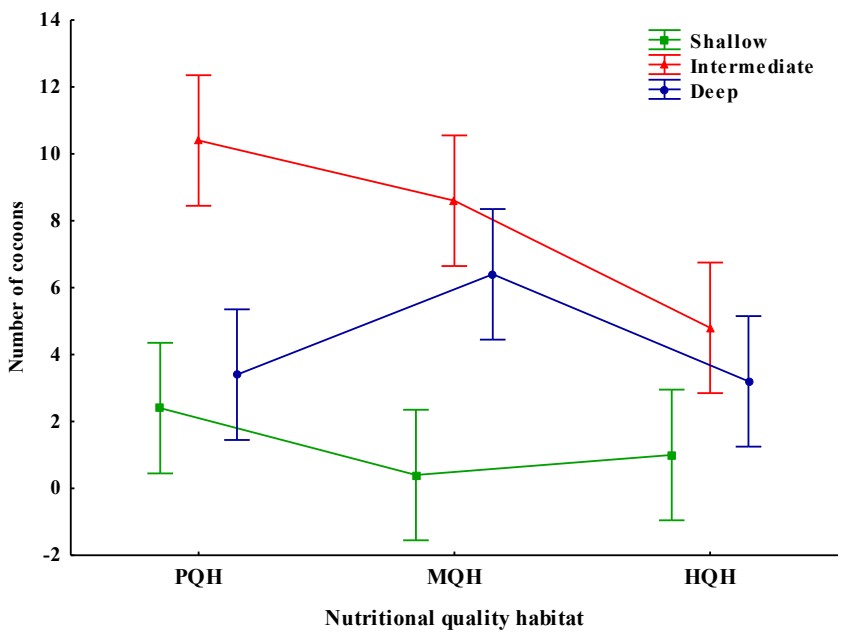

**Figure 3** Interaction between the depth and nutritional quality of the soil on nest construction in the tropical endogeic earthworm *Pontoscolex corethrurus.* Soil depth: Shallow = 0–9 cm, Intermediate = 10–18 cm, Deep = 19–27 cm. Nutritional quality of the habitat: PQH = soil only, MQH = soil + grass, HQH = soil + legume. Vertical lines indicate 95% confidence intervals.

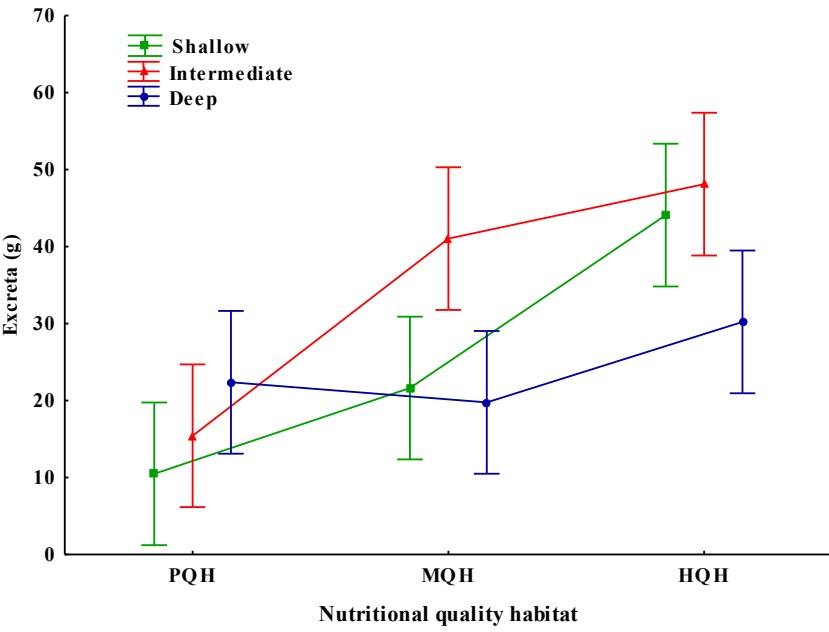

**Figure 4** Interaction between the depth and nutritional quality of the soil on the production of internal excreta in the tropical endogeic earthworm *Pontoscolex corethrurus.* Soil depth: Shallow = 0–9 cm, Intermediate = 10–18 cm, Deep = 19–27 cm. Nutritional quality of the habitat: PQH = soil only, MQH = soil + grass, HQH = soil + legume. Vertical lines indicate 95% confidence intervals.

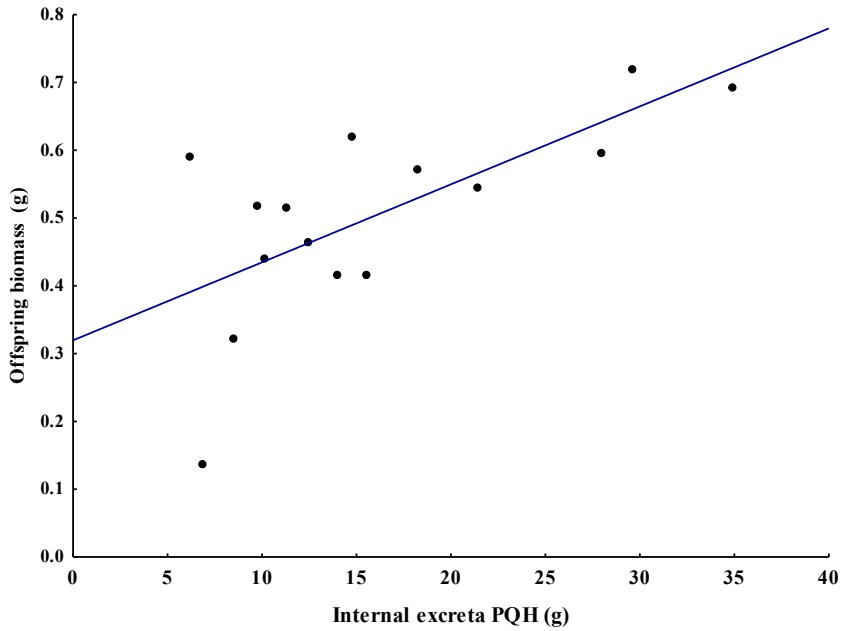

**Figure 5** **Spatial association between biomass offspring and internal excretas (treatments S, N = 15) of the tropical endogeic earthworm _Pontoscolex corethrurus._** PQH = only Soil. The line in fitted with a linear regression.

## DISCUSSION

Low quality of the habitat can drive the evolution of parental care, and this can vary as a function of the distribution, abundance, persistence and quality of different food resources (_Tallamy & Wood, 1986_; _Mori & Chiba, 2009_; _Smiseth, Kölliker & Royle, 2012_). _Pontoscolex corethrurus_ constructed chambers or nests similar to those recorded in previous studies (_Ortiz-Ceballos & Fragoso, 2006_; _Ortiz-Ceballos, Hernández-García & Galindo-González, 2009_; _Buch et al., 2011_). Contrary to our prediction, a higher quantity of nests were constructed and deposited at an intermediate depth in the PQH and MQH treatments, which corresponds to the lower and medium quality, respectively. However, more internal excreta where deposited at the HQH-Intermediate treatment, which corresponds to the high quality environment.

The inspection and selection of potential sites for oviposition is one of the most important patterns of behaviour in animals (_Lentfer et al., 2011_; _Smiseth, Kölliker & Royle, 2012_). The selection of nest sites may increase offspring survival by choosing adequate abiotic factors such as soil moisture, temperature, and soil depth associated with the gas exchange ($O_2$ and $CO_2$) required for incubation. For example, in the savannah of Colombia "los Llanos" the mean depth at which cocoons were laid for the native earthworm _Glossodrilus sikuani_ was 8.8 in the original savannah and 12.4 cm in an introduced pasture (_Jiménez, 1999_; _Jiménez et al., 1999_). Our results indicate that the parental behavior of _P. corethrurus_ varied significantly with soil depth and habitat quality, a higher amount of cocoons were placed at an intermediate depth (10–18 cm) in the soil with poor and medium quality (PQH and

MQH treatments). We suggest this could be a form of diversified bet-hedging strategy (do not place all cocoons in one basket) when faced with changes in the abundance, quality or predictability of food resources (*Olofsson, Ripa & Jonzén, 2009*; *Nevoux et al., 2010*; *Simons, 2011*). Bet-hedging theory addresses how individuals should optimize their fitness in a variable and unpredictable environment (*Olofsson, Ripa & Jonzén, 2009*; *Nevoux et al., 2010*; *Simons, 2011*). It seems counter intuitive that at the sites with high quality habitat (HQH) there were less cocoons produced than at the poor and medium quality. However, at HQH there could be increased risk of predation if these soil qualities attract other soil organisms.

One simple form of parental care is to bury eggs in a substrate (*Smiseth, Kölliker & Royle, 2012*). For example, *L. terrestris*, covers its eggs with its own excreta (*Ramisch & Graff, 1985*; *Grigoropoulou, Butt & Lowe, 2008*). There are more elaborate forms of nest construction using materials found in the environment (natural or processed), or the parents can use self-produced materials such as mucus or silk, among others (*Jeanne, 1996*; *Mori & Chiba, 2009*; *Smiseth, Kölliker & Royle, 2012*; *Furuichi & Kasuya, 2015*). As observed by *Ortiz-Ceballos, Hernández-García & Galindo-González (2009)*, and *Buch et al. (2011)* both in the field collections and the laboratory, here we found that *P. corethrurus* uses soil and mucus to construct nests with its mouth that are similar to those constructed in diapause by *Millsonia anomala* (*Blanchart et al., 1997*) and *Martiodrilus carimaguensis* (*Jiménez et al., 2000*). It has been suggested that nest architecture has evolved for multiple uses where the exterior layer acts to conceal the nest from predators and protect it from rain while the internal layer isolates it from temperature extremes, flooding, desiccation and hypoxia (*Mori & Chiba, 2009*; *Smiseth, Kölliker & Royle, 2012*; *Kingsbury et al., 2015*). The nests constructed here could be a form of parental care to protect the cocoons from abiotic (reducing water loss and improving gas exchange) and biotic (predators) threats (*Ortiz-Ceballos, Hernández-García & Galindo-González, 2009*), since the interior layer comprises a compacted wall formed by small soil particles bound together by mucus produced by the earthworm, while the exterior layer acts to disguise the presence of the nest. Furthermore, the cocoons within the nest are suspended from a transparent layer of mucus (*Ortiz-Ceballos, Hernández-García & Galindo-González, 2009*; *Buch et al., 2011*). This is probably associated with sanitation (antimicrobial properties), and can be found in epigeic earthworms (*Eisenia fetida*), beetles (*Dendroctonus frontalis* and *Nicrophorus vespilloides*), hyperiid amphipods (genus *Phronima*), the European beewolf (*Philanthus triangulum*), ants and termites (*Currie, 2001*; *Kaltenpoth et al., 2005*; *Muller et al., 2005*; *Hirose, Aoki & Nishikawa, 2005*; *Aruna et al., 2008*; *Rozen, Engelmoe & Smiseth, 2008*; *Scott et al., 2008*; *Smiseth, Kölliker & Royle, 2012*).

Another evolutionary characteristic of parental care is improvement of the food quality (providing excreta of greater quality and of a particle size suitable for consumption) available to the offspring in order to sustain growth and reduce mortality and the time necessary for development (*Mori & Chiba, 2009*; *Gardner & Smiseth, 2011*). For example, larvae of the beetle *Figulus binobulus* feed on excreta that are rich in nitrogen and sawdust (*Mori & Chiba, 2009*). In xylophagous insects, the larvae feed on excreta rich in proteins produced by their parents (*Ento, Araya & Kudo, 2008*). Soil is a difficult environment in which it is hard to find plant material with nutritional value (*Bonkowski, Griffiths & Ritz, 2000*). Thus, providing highly nutritious excreta for offspring may increase their
biomass and survival. Earthworms prefer leaf litter with high N content (*Hendriksen, 1990*; *Bonkowski, Griffiths & Ritz, 2000*). This may explain why the soil with the highest nutritional quality (those mixed with the legume) presented a lower number of nests but produced excreta with high nutritional value as a source of food for their offspring. This leads us to suppose these sites are essential for the reproductive activity (nutrition) of *P. corethrurus* (*Lee, 1985*; *García & Fragoso, 2003*; *Ortiz-Ceballos et al., 2005*). Excreta are thought to contain nutritional resources with a high N and P content, they contain a water-soluble mixture of low molecular weight carbohydrates, aminoacids, glycosides and a glycoproteins, humic substances (endowed with hormone-like activity), and low C:N content. They can cause priming effects by stimulating microbial activity (*Elliot, Knight & Anderson, 1991*; *Tiwari & Mishra, 1993*; *Nardi et al., 1994*; *Decaëns et al., 1999*; *Musculo et al., 1999*; *Trigo et al., 1999*; *Whalen, Parmelee & Subler, 2000*; *Salmon, 2001*; *Schönholzer et al., 2002*; *Ihssen et al., 2003*; *Egert et al., 2004*; *Furlong et al., 2002*; *Drake & Horn, 2007*; *Oleynik & Byzov, 2008*; *Bityutskii, Maiorov & Orlova, 2012*; *Lipiec et al., 2015*). Our results show that the interaction between habitat quality and soil depth had a significant influence on the production of internal excreta, a higher amount of internal excreta were placed at an intermediate depth (10–18 cm) in the soil with high quality (HQH-Intermediate treatment).

The vertical and horizontal movement of ingested and transported materials within the soil (earthworm bioturbation: translocation of soil materials via excreta) is most apparent when it involves the deposition of excreta at the surface (*García & Fragoso, 2002*; *Ortiz-Ceballos, Hernández-García & Galindo-González, 2009*). However, our results suggest that as a form of diversified bet-hedging, *P. corethrurus* could have ingested high quality material, at the HQH treatment, concentrating it in its excreta and transporting it to the low quality sites. Similar behaviour has been documented by *García & Fragoso (2002)* where *P. corethrurus* transported higher rates of excreta from sites with high organic material to sites with low quality (mineral soil).

Excreta were deposited close to the nests in a similar manner to that reported in a previous study (*Ortiz-Ceballos, Hernández-García & Galindo-González, 2009*), in contrast to *L. terrestris*, which covers its cocoons with its own excreta (*Ramisch & Graff, 1985*; *Grigoropoulou, Butt & Lowe, 2008*). After hatching, the offspring consume the internal excreta, perhaps to survive (*Ortiz-Ceballos, Hernández-García & Galindo-González, 2009*). The internal excreta is characterized by fine soil particles (the mouth of the offspring is not adapted to consume large soil particles), organic matter, humic substances, nitrogen and microorganisms (*Tiwari & Mishra, 1993*; *Devliegher & Verstraete, 1997*; *Trigo et al., 1999*; *Bonkowski, Griffiths & Ritz, 2000*; *Lowe & Butt, 2002*; *Curry & Schmidt, 2007*; *Khomyakov et al., 2007*; *Mariani et al., 2007*; *Mori & Chiba, 2009*). In this way, the offspring obtain nutrients suitable for their growth and development and at the low and medium habitat quality (PQH and MQH) they had a significant positive association with juvenile biomass. This suggests that *P. corethrurus* as an additional form of parental care provides food (excreta with nutrients and humic substances). Reproductive investment is not only in cocoon production (4.56% of their weight), but also in nest site construction and excreta transportation. We found that earthworms selected nest sites and produce more offspring

in poor and medium quality condition perhaps as a bet-hedging strategy, while depositing more excreta in high-quality habitats.

## CONCLUSIONS

As part of its reproductive activity, *P. corethrurus* could have two reproductive strategies that act as diversified bet-hedging (do not place all cocoons in one basket) in unpredictable environments of the soil; one to build a higher amount of nests in low and medium quality habitats; and the other to produce more internal excreta as a form of parental care in high quality habitats. Cocoons are placed in nests and additionally excreta are deposited as a source of food for the offspring. Parental care in the form of internal excreta may be energetically expensive, but may be particularly important in poor and medium quality habitats where offspring biomass increased with internal excreta. Further research is necessary to determine whether species of different ecological categories also provide parental care for their offspring.

## ACKNOWLEDGEMENTS

We thank Mario Favila, Carlos Fragoso, José A. García-Pérez and Roberto Munguía-Steyer for helpful discussion; Carolina Cruz González, Sué Olive Vázquez Rodríguez, Narciso Acosta Medel and Diana Ortiz Gamino for technical assistance. We also thank two anonymous reviewers.

### Funding

Funding was provided by a CONACyT Ciencia Básica Grant (CB-2007-01/83600). The funders had no role in study design, data collection and analysis, decision to publish, or preparation of the manuscript.

### Grant Disclosures

The following grant information was disclosed by the authors:
CONACyT Ciencia Básica Grant: CB-2007-01/83600.

### Competing Interests

The authors declare there are no competing interests.

### Author Contributions

- Angel I. Ortiz-Ceballos conceived and designed the experiments, performed the experiments, analyzed the data, contributed reagents/materials/analysis tools, wrote the paper, prepared figures and/or tables, reviewed drafts of the paper.
- Diana Pérez-Staples performed the experiments, wrote the paper, prepared figures and/or tables, reviewed drafts of the paper.
- Paulino Pérez-Rodríguez analyzed the data, wrote the paper, prepared figures and/or tables, reviewed drafts of the paper.

## Data Availability

The raw data has been supplied as Data S1.

## Supplemental Information

Supplemental information for this article can be found online at http://dx.doi.org/10.7717/peerj.2032#supplemental-information.

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
