# Peer review of "Nest site selection and nutritional provision through excreta: a form of parental care in a tropical endogeic earthworm"

_PeerJ, doi:10.7717/peerj.2032_

## Round 0.1 · original submission · Major Revisions

· Academic Editor

Major Revisions

Dear authors,

Please revise the manuscript carefully taking into consideration the reviewers comments and send it back to the journal for final decision.

Reviewer 1 ·

Basic reporting

This is an interesting study, and it is clear to read, but I think that : the abstract, Introduction and discussion requires some work, thats why I am suggesting major revisions for the paper,
in the abstract: i suggest to change the explanation of : P. corethrurus selects the sites according too... in: P. corethrurus produces nests in those soils with low nutritional quality
Intro & background to show context. It is well written, with good backgroung, but the hypothesis, it is not really clear, they explain that the parental care among the earthworms is an strategy in order to insure the progeny, and then they hypothesize that P. corethrurus can select the place where they will put their nests an cocoons, I would formulate the hypothesis more in relation of the “quality” of the habitat, places with more quality will require less nest, less parental care, and places with low “quality” will demand more nests, and more parental care, as a consequence of adaptation and assuring the survival of the progeny.
And then this hypothesis will justify the use of treatments: control (only soil, “poor quality habitat”), Grass+Soil (medium quality habitat), Leguminose +Soil (high quality habitat)
In the introduction the authors have to add a small paragraph indicating how is the production of cocoons on the field, how many cocoons of P. corethrurus have been found in natural sites (rich in nutrients,) and how many cocoons of P. corethrurus have been found in managed sites (perturbed areas).
-Literature well referenced & relevant. Yes, but I suggest to inform about the production of cocoons of P. corethrurus along different systems (natural, managed) and areas, P. corethrurus is an exotic earthworm in Mexico, it is native from South America, the authors must indicate how is the variation of P. corethrurus cocoons production also with field information.
-Structure conforms to PeerJ standard, discipline norm, or improved for clarity. Abstract, introduction and Material and methods must be improved
-Figures are relevant, high quality, well labelled & described. Yes
-Raw data supplied (See PeerJ policy). yes

Experimental design

Its an innovative study, but some details have to be given, please:
Material and methods. The authors can indicate please why they didn’t collect the worms from Tamulte de las Sabanas?, normally when you breed earthworms, you collect earthworms and soil from the same place (because of the gut microflora adaptation)
In material and methods, they authors talk about the quality of the food, ok!, they have to address this matter also in the introduction.
Indicate which is the field capacity of the soil of the experiment
Have you recorded the moisture along the different levels of the terrarium?, please inform,
Please relocate the paragraph of the terrariums inside the Experimental Set up, also indicate here that due that P. coretrurus is partenogenetic, one earthworm per terrarium was put.
Can the authors provide with a diagram or photo of the terrarium?, it will be very useful for better understanding.
Can the authors say why they chose that temperature for incubating the terrarium? If it is based on references (with this temperature P. corethrurus produce more cocoons…, indicate it please), or is the soil temperature in Huimanguillo??? (where worms were collected?, please indicate) or both?
Can authors indicate how moisture was measured?
In data analysis, please indicate that the Anova was performed in order to record differences among treatments

Validity of the findings

Figures are good, they have good quality, but authors have to be more precaucious, there is more excreta production in L treatments, and more cocoons production in S treatments, there was a strong correlation between offspring biomass and internal excreta, but then? what is the role of the nests? it is not clear explain in discussion,
In results, line 208, indicate that no differences on biomass… were found among treatments
In discussion, Line 285, is true but recorded in different treatments (excreta and cocoons), maybe the Intermediate location was used by the worms, because there, there was less light influence?, or the moisture was different?
The authors in discussion have to explain more about the use of resources by P. corethrurus, and how this derives into the production of nests, and they have to inform it this has been observed on field?.
The authors have to explain what is the relevance on the distribution of P. corethrurus energy the fact of protecting or not protecting their cocoons.
The conclusion is not quite accurate, they found more excreta in the leguminose treatment, but more nests in the soil and grass treatments, I suggest that they have to reformulate more the discussion, indicating that indeed the nest production is a strategy that P. corethrurus has developed in order to protect the cocoons in habitat with not high quality.

Additional comments

It is an interesting work, really innovative, but information of P.corethrurus cocoon production on the field is important to be added, how is the production of cocoons in natural and managed sites, with low and high quality sites.

Annotated reviews are not available for download in order to protect the identity of reviewers who chose to remain anonymous.

·

Basic reporting

This article would benefit from a more detailed description of the performed procedures. Also a picture of the terrarium at the beginning and the end of the experiments could help to get a better impression of what has been observed.

I'd like to remark that the term 'tridimensional terrarium' (line 29 and 105) is not very meaningful for such a flat vessel. In previous papers this kind of containers was referred to as ‘planar cuvettes’ (Evans 1947) or ‘2D terraria’ (Capowiez 2000).

Experimental design

Obviously, the experimental arrangement did not cover all possible treatment variations in full permutation. The variants S-Shallow/G- Intermediate/L-Deep, L-Shallow/S-Intermediate/G-Deep and G-Shallow/L-Intermediate/S-Deep were not tested. This would be relevant, if not only the effect of food quality in different depth, but also the influence of adjacent layers should be considered.

I wonder to what extent the soil was compacted by placing it into the cuvettes. Nothing is said about the movements of the worms within and between the layers. Did they construct permanent galleries and were these distributed evenly or concentrated in preferred layers? How much bioturbation took place resulting in a mixture of the layers?

Please describe how you separated casts (internal excreta) from unspoiled soil at the end of the experiment. It is difficult to understand, how you could assign the biomass of juveniles to the amount of casts in the particular layer. Didn't the juveniles move across the layer boundaries?

Validity of the findings

I miss a plausible explanation for the coincidence, on the one hand, of the smaller number of nests in treatment L (soil + legume) compared to treatments S and G in the intermediate soil depth layer and, on the other hand, the higher amount of internal exceta in that layer. If you see this in relationship to the final quote “perhaps the most distinguishing feature of earthworms is their propensity to consume their house”, it should be presented more comprehensible.

---

## Round 0.2 · accepted · Accept

· Academic Editor

Accept

Thank you for your careful revisions, I am happy to Accept the paper.